# A general orientation distribution function for clay-rich media

Thomas Dabat [1]*, Fabien Hubert[1]*, Erwan Paineau [2], Pascale Launois [2], Claude Laforest[1], Brian Grégoire [1], Baptiste Dazas[1], Emmanuel Tertre[1], Alfred Delville[3] & Eric Ferrage [1]*

The role of the preferential orientation of clay platelets on the properties of a wide range of natural and engineered clay-rich media is well established. However, a reference function for describing the orientation of clay platelets in these different materials is still lacking. Here, we conducted a systematic study on a large panel of laboratory-made samples, including different clay types or preparation methods. By analyzing the orientation distribution functions obtained by X-ray scattering, we identified a unique signature for the preferred orientation of clay platelets and determined an associated reference orientation function using the maximum-entropy method. This new orientation distribution function is validated for a large set of engineered clay materials and for representative natural clay-rich rocks. This reference function has many potential applications where consideration of preferred orientation is required, including better long-term prediction of water and solute transfer or improved designs for new generations of innovative materials.

[1] IC2MP-Hydrasa, UMR 7285 CNRS, Université de Poitiers, 86022 Poitiers, France. [2] Laboratoire de Physique des Solides, UMR 8502 CNRS, Université Paris Saclay, bât. 510, 91405 Orsay, France. [3] ICMN, UMR 7374 CNRS, Université d'Orléans, 45071 Orléans, France. *email: thomas.dabat@univ-poitiers.fr; fabien.hubert@univ-poitiers.fr; eric.ferrage@univ-poitiers.fr

Clay minerals are one of the fundamental constituents of numerous terrestrial and marine environments on Earth and are systematically present in soils and sedimentary rocks[1]. The sub-micrometric particle sizes of clay particles most often account for a major proportion of the specific surface area of the whole material, leading to strong interactions with organic macromolecules or dissolved cations and anions. Owing to their lamellar shape, the preferred orientation of clay platelets gives rise to anisotropy in the morphology of the pore network[2]. Both effects have considerable influence on retention and anisotropic transfer of water and solutes in clay-rich media, causing clay minerals to play a pivotal role in the exploration and management of critical resources for human beings.

In sedimentary rocks such as mudrocks or shales, clay minerals are common markers for paleoenvironmental reconstructions[3,4] and the preferred orientation of clay platelets associated with nanometer-sized pores provides low permeability values to the whole clay-rich material. This latter process has a major effect on the development of oil, gas, water reservoirs, and is at the origin of the consideration of these clay-rich media as a natural barrier for $CO_2$ capture and nuclear waste storage[5–7]. The omnipresence of clay minerals in soil environments is also involved in the development of thin crusts at the top surface. The significant preferred alignment of clay platelets in this environment[8] and the low permeability of the material exert a key effect on soil erosion and directional dependence of water and pollutant migration[9]. Clay minerals are also used in engineered materials given their low cost, abundance and the availability of natural deposits. Tuning the desirable fluid and gas barrier efficiency of these engineered materials through orientation control of clay platelets offers a variety of innovative sealing solutions ranging from rigid gaskets for high-temperature industrial applications[10] to flexible and transparent films for packaging applications in food and electronic industries[11–14].

Although anisotropic features in clay platelet alignment have demonstrated their importance in numerous natural or engineered materials, a general function predicting preferred orientation for clay platelets is still lacking. Preferred orientation is generally studied through the analysis of the orientation distribution (OD) of crystallites by diffraction methods[8,15–20]. The extent of anisotropy in clay platelet orientation is then most often illustrated by global descriptors of the experimental orientation distribution function (ODF; e.g., the width of the distribution or its maximum value expressed as a multiple of random distribution)[8,15–18,21,22] or by an adjustable spreading parameter of conventional distribution functions (Gaussian, Bingham, etc.[18,21,23,24]) fitted to the experimental data. Though providing a more complete description of the shape of the OD, these latter analyses remain targeted towards a limited number of clay minerals and preparation methods or are restricted to a narrow range of anisotropy degrees. To date, no conventional distribution function has been shown to be applicable for a large set of data. This lack of a reference function describing OD for clay minerals thus leads to significant uncertainties in the prediction of long-term transfers of critical resources in compacted clay-rich systems[5,25] or for driving the design of high-performance engineered materials[26].

Herein, our objective is to fill this crucial gap by providing a general reference function of clay platelet alignment in a large number of clay-rich engineered and natural materials. We first develop a large set of laboratory clay-based engineered materials covering the most characteristic clay types (kaolinite, mica, chlorite, vermiculite, and smectite) and a broad panel of preparation methods (compaction, sedimentation, and centrifugation). Experimental orientation distributions of clay mineral platelets are then determined by X-ray scattering (XRS)

measurements. Through detailed analysis of the different momenta of the experimental data, we unravel, for the first time, the existence of a specific orientation signature independent of the type of clay platelet or preparation method. We also demonstrate that none of the conventional distribution models used so far allows for the reproduction of the overall set of measured ODFs. Based on the maximum-entropy method, we propose a new function that allows a complete description of the experimental OD for clay platelets throughout the whole range of anisotropy degrees. Finally, the efficiency of this new distribution function in modeling the preferred orientation of clay platelets is illustrated for a representative set of engineered samples, as well as for three natural systems, i.e., a shale, a soil crust, and a schist, representative of the main different clay-rich environments on Earth's surface or subsurface.

## Results

**Basics of the orientation distribution function and its determination from X-ray scattering experiments.** The orientation of a clay platelet in an orthogonal framework $Oxyz$ is determined by the orientation of its normal $Oz'$, which is defined by its spherical coordinates $\theta$ and $\varphi$ (Fig. 1a). Clay systems are transverse isotropic media with the $z$-axis as the unique symmetry axis[15]. The orientation distribution function $f(\theta, \varphi)$ thus only depends on the angle $\theta$ where:

$$f(\theta) \geq 0 \qquad (1)$$

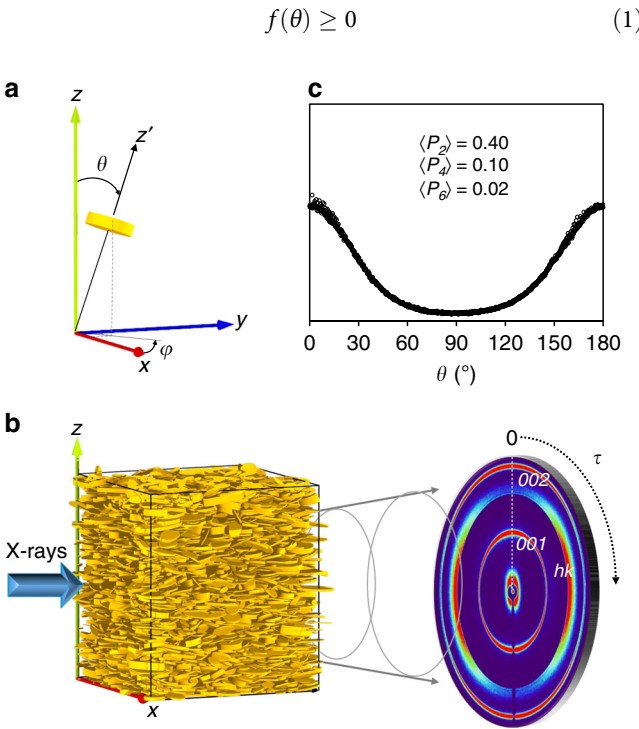

**Fig. 1** Experimental measurement of the orientation distribution function. **a** Spherical coordinates ($\theta$, $\varphi$) defining the orientation of an individual clay platelet (in yellow) with respect to the laboratory frame. **b** Schematic representation of an X-ray scattering experiment. The sample is studied in transmission, and scattered intensities are recorded on a planar detector where the color scale ranges from dark blue for the smallest intensity values to red for the largest values. The image here corresponds to a kaolinite sample. The rings corresponding to *00l* Bragg reflections and *hk* reflections (i.e., *02* and *11* reflections) have maximum intensities along the vertical *z* direction and the horizontal *x* direction, respectively. **c** A typical experimental ODF deduced from the angular scan of the *001* diffraction ring using Eq. (4), with the calculated $\langle P_2 \rangle$, $\langle P_4 \rangle$, and $\langle P_6 \rangle$ order parameter values from Eq. (10).

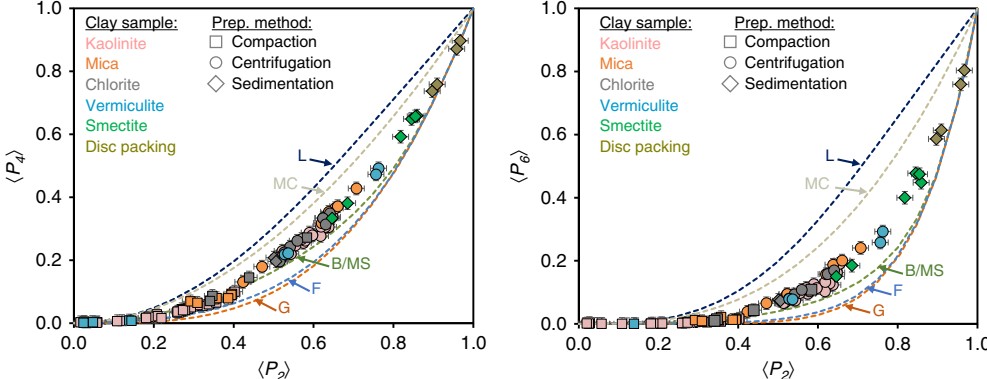

**Fig. 2** Relationships between the experimental $\langle P_2 \rangle$, $\langle P_4 \rangle$, and $\langle P_6 \rangle$ order parameters. Momenta of the experimental ODF for kaolinite, mica, chlorite, vermiculite, smectite, and artificial clay samples made of disc packings[34] are shown in pink, orange, grey, blue, green, and dark green, respectively. Sample preparation based on compaction, centrifugation, and sedimentation methods are depicted by squares, circles, and diamonds, respectively. Experimental order parameters are compared with calculated parameters, with conventional Gaussian (G), Lorentzian (L), Fisher (F), Bingham/Maier-Saupe (B/MS), and mechanical compaction (MC) distribution models shown as orange, dark blue, light blue, green, and gray dotted lines, respectively.

$$f(\theta) = f(\pi - \theta) \qquad (2)$$

$$\int_0^\pi f(\theta)\sin(\theta)\mathrm{d}\theta = 1 \qquad (3)$$

Any distribution function is positive and can be normalized to unity[27], which yields Eqs. (1) and (3), respectively. The platelet orientation is equivalently characterized by a unit vector parallel to $Oz'$ and by its opposite, defined by the spherical coordinates $(\theta, \varphi)$ and $(\pi - \theta, \varphi + \pi)$ from Eq. (2). In the XRS experiments performed here, scattered intensities are recorded on a planar detector (Fig. 1b). Due to the preferred orientations of the platelets, the intensity of the 001 diffraction ring is modulated angularly[28]. Assuming that the angle $\tau$ on the detector is equal to the spherical coordinate $\theta$ (see Supplementary Note 1, Supplementary Figs. 1–3), the ODF $f(\theta)$ (Fig. 1c) can be fully determined from the angular modulation of intensity $I$ along the 001 diffraction ring:

$$f(\theta) = \frac{I(\theta)}{\int_0^\pi I(\theta)\sin(\theta)\mathrm{d}\theta} \qquad (4)$$

**Momenta analysis of the experimentally determined orientation distribution functions for clay platelets.** The ODF $f(\theta)$ can be developed as an infinite series of Legendre's polynomial functions $P_l(\cos\theta)$ as:

$$f(\theta) = \sum_{l=0}^{\infty} \frac{2l+1}{2} \langle P_l(\cos\theta) \rangle P_l(\cos\theta) \qquad (5)$$

The sum can be restricted to even values of $l$ as Eq. (2) leads to null values of odd terms of the series. The first Legendre polynomials in the expansion are expressed as follows:

$$P_0(\cos\theta) = 1 \qquad (6)$$

$$P_2(\cos\theta) = \frac{1}{2}\left(3\cos^2\theta - 1\right) \qquad (7)$$

$$P_4(\cos\theta) = \frac{1}{8}\left(35\cos^4\theta - 30\cos^2\theta + 3\right) \qquad (8)$$

$$P_6(\cos\theta) = \frac{1}{16}\left(231\cos^6\theta - 315\cos^4\theta + 105\cos^2\theta - 5\right) \qquad (9)$$

The different momenta of the distribution function, also defined as order parameters, are:

$$\langle P_l \rangle = \langle P_l(\cos\theta) \rangle = \int_0^\pi P_l(\cos\theta).f(\theta)\sin(\theta)\mathrm{d}\theta \qquad (10)$$

Analysis of orientation anisotropy most often relies on the value of the second momentum $\langle P_2 \rangle$. This parameter is known as $H$, the Hermans parameter in polymer and composites sciences[29], or $S$, the nematic order parameter in colloid science[30,31]. This parameter has a value of 0 for isotropic organization and 1 for perfectly aligned particles. However, $\langle P_2 \rangle$ also has a null value when all particles are oriented at the magic angle $\theta = acos(1/\sqrt{3}) \simeq 54.7°$. This specificity illustrates the limit of using only the $\langle P_2 \rangle$ order parameter to characterize the anisotropy. The actual fingerprint of the shape of the ODF and the whole set of momenta $\langle P_l \rangle$ are accessible from XRS analysis according to Eqs. (4) and (10). This contrasts with other vibrational techniques such as infrared or Raman spectroscopies, which provide only $\langle P_2 \rangle$ or $\langle P_2 \rangle$ and $\langle P_4 \rangle$ momenta[32,33], respectively, thus limiting the number of Legendre polynomials to the fourth rank order in Eq. (5).

In an effort to obtain information that is as general as possible on the ODF for clay-rich media, a large set of different clay types (kaolinite, mica, chlorite, vermiculite, and smectite) and preparation methods (compaction, sedimentation, and centrifugation) are considered here. These different clay types represent the most ubiquitous clay minerals on the Earth's subsurface[1,3]. The three preparation protocols are considered here to be representative of the main different formation mechanisms of clay-rich natural or engineered materials. These protocols include the compaction process in dried conditions (i.e., compaction method), the settling of clay particles in suspension (i.e., sedimentation method), and a coupled sedimentation-compaction method in water saturated conditions mimicking the early diagenesis process of minerals (i.e., centrifugation method). A total of 100 XRS analyses, complemented by recent anisotropy measurements on four sedimented artificial clay samples[34], provide a large experimental dataset for the analysis of ODF and a wide range of anisotropy degrees (i.e., order parameter values). Among the whole set of $\langle P_l \rangle$ order parameters accessible by XRS analysis, Fig. 2 shows the first momenta $\langle P_4 \rangle$ and $\langle P_6 \rangle$ as a function of $\langle P_2 \rangle$ for all samples

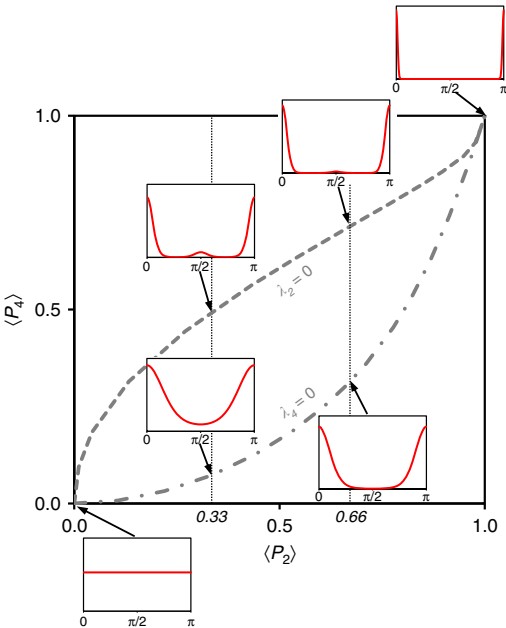

**Fig. 3** The $\langle P_4 \rangle - \langle P_2 \rangle$ order parameters interrelations based on maximum-entropy theory. According to MEM theory, each pair $(\lambda_2, \lambda_4)$ defines a full set of $\langle P_l \rangle$ momenta and thus a given shape of the ODF. Here, selected representative ODFs (calculated from Eq. (18)) are shown in inserts as red lines in the case of $\lambda_2 = 0$ or $\lambda_4 = 0$ for $\langle P_2 \rangle$ values of 0.00, 0.33, 0.66, and ~1.

investigated (with ODF determined from Eq. (4) and order parameters from Eq. (10)) to assess a potential common signature of ODFs. Regardless of the nature of the clay mineral (i.e., kaolinite, mica, chlorite, vermiculite, and smectite) or the preparation method (i.e., compaction, sedimentation, or centrifugation), univocal relationships are evident for the three first momenta of the ODF, revealing a specific signature for the preferred orientation of clay platelets.

For simplicity, experimental OD profiles are commonly fitted in the literature with conventional distribution models. For clay minerals, the most commonly used distribution functions are the Gaussian[18,21,35,36] $(f_G(\theta))$, Lorentzian[37] $(f_L(\theta))$, Fisher[24,38] $(f_F(\theta))$, Bingham or Maier-Saupe[23,24,38] $(f_{B/MS}(\theta))$, and the mechanical compaction models[24,39] $(f_{MC}(\theta))$, taking the following form:

$$f_G(\theta) = k_G(m) \exp\left(-\frac{\theta^2}{m}\right) \tag{11}$$

$$f_L(\theta) = k_L(m) \frac{m}{\theta^2 + m} \tag{12}$$

$$f_F(\theta) = k_F(m) \exp(m \cos\theta) \tag{13}$$

$$f_{B/MS}(\theta) = k_{B/MS}(m) \exp(m \cos^2\theta) \tag{14}$$

$$f_{MC}(\theta) = k_{MC}(m) \frac{m}{(\cos^2\theta + m \sin^2\theta)^{3/2}} \tag{15}$$

Note that to satisfy the symmetry of the ODF stated in Eq. (2), the Gaussian, Lorentzian and Fisher functions $f_i(\theta)$ (with $i = G, L, F$) are summed up with $f_i(\pi - \theta)$[24]. The prefactor $k_i(m)$, with $i = G, L, F, B/MS$, or MC, is a normalization constant to satisfy Eq. (3), and $m$ is the parameter governing the ODF spread.

For each function and by varying the parameter $m$, we calculated order parameters for $l = 2, 4,$ and 6. The calculated $\langle P_4 \rangle - \langle P_2 \rangle$ and $\langle P_6 \rangle - \langle P_2 \rangle$ curves are reported in Fig. 2. It is clear that none of the standard models are able to correctly reproduce

our experimental dataset. The Lorentzian and mechanical compaction distributions indeed clearly overestimate the $\langle P_4 \rangle$ and $\langle P_6 \rangle$ order parameters through the entire range of $\langle P_2 \rangle$ values. The Gaussian and Fisher models provide good agreement with experimental data for small and large $\langle P_2 \rangle$ values, i.e., below 0.2 and above 0.9, but they significantly underestimate the $\langle P_4 \rangle$ and $P_6$ order parameter values in the intermediate range of $\langle P_2 \rangle$ values. Finally, the Bingham or Maier-Saupe distribution model provides the closest agreement with our data but still fails to account for $\langle P_4 \rangle$ and $\langle P_6 \rangle$ contributions in the $0.4 < \langle P_2 \rangle < 0.9$ region. The present results clearly demonstrate the limitations of conventional distribution models in reproducing the full signature of the clay mineral ODFs across the whole range of anisotropy degrees.

**Determination of a general orientation distribution function based on the maximum-entropy method.** To describe the entire ODF for clay-rich media, an alternative to the limitation of existing distribution models is to consider the maximum-entropy method (MEM) for reconstructing the full orientation function. The MEM relies on information theory (in statistics) and provides the most probable ODF that is consistent with a set of few known $\langle P_l \rangle$ values[27,32,40,41]. One estimates the equilibrium ODF and thus the entire range of $\langle P_l \rangle$ parameters by maximizing the information entropy defined as:

$$S(f(\theta)) = -\int_0^\pi f(\theta) \ln(f(\theta)) \sin(\theta) d\theta \tag{16}$$

The ODF can then be written as:

$$f_{MEM}(\theta) = k_{MEM} \exp\left(\sum_{i=2}^n \lambda_i P_i(\cos\theta)\right) \tag{17}$$

where $n$ is the index of the higher order parameter used as a constraint (the sum over $i$ in Eq. (17) runs on even numbers between 2 and $n$), $\lambda_i$ represents the Lagrange multipliers, and $k_{MEM}(\lambda_2, \ldots, \lambda_n)$ is the normalization constant that satisfies the relation in Eq. (3). The index 'MEM' refers to the maximum-entropy method. The MEM approach has been proven to be very efficient when only a few order parameters are known[40,41]. Indeed such a reduced knowledge yields Eq. (5) inoperable for high degree of anisotropy. The situation differs here as XRS analysis provides the full $f(\theta)$ ODF and thus the entire set of order parameters. The objective in the present work is to beneficiate from MEM theory to derive a reference distribution function based on a minimum number of parameters (i.e., $\lambda_i$ coefficients) so that the whole set of $\langle P_l \rangle$ momenta would be automatically accounted for. By limiting the reconstruction of the $f_{MEM}(\theta)$ function to the two first $\lambda_i$ parameters (i.e., $n = 4$), the ODF takes the following form:

$$f_{MEM}(\theta) = k_{MEM}(\lambda_2, \lambda_4) \exp(\lambda_2 P_2(\cos\theta) + \lambda_4 P_4(\cos\theta)) \tag{18}$$

with Lagrange multipliers $\lambda_2$ and $\lambda_4$ calculated to satisfy the experimental constraints:

$$\int_0^\pi P_2(\cos\theta).f_{MEM}(\theta) \sin(\theta) d\theta = \langle P_2 \rangle \tag{19}$$

$$\int_0^\pi P_4(\cos\theta).f_{MEM}(\theta) \sin(\theta) d\theta = \langle P_4 \rangle \tag{20}$$

According to MEM theory, each pair $(\lambda_2, \lambda_4)$ defines a full set of $\langle P_l \rangle$ momenta. Figure 3 reports the interrelation between Lagrange multipliers and the first $\langle P_2 \rangle$ or $\langle P_4 \rangle$ momenta and illustrates the associated shapes of ODFs[41].

The isotropic configuration of particles is represented by a horizontal line associated with $\lambda_2 = \lambda_4 = 0$ and leads to $\langle P_2 \rangle =$

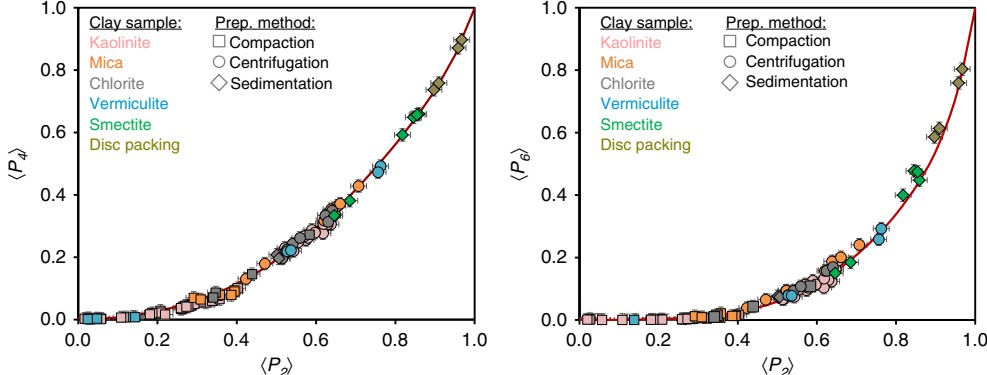

**Fig. 4** Determination of the general orientation distribution function for clay minerals. Experimental momenta $\langle P_4 \rangle - \langle P_2 \rangle$ or $\langle P_6 \rangle - \langle P_2 \rangle$ relationships (same notations as Fig. 2) are compared to calculated momenta using the general ODF for clay minerals ($f_{CM}(\theta)$ using Eq. (21)) (red line) based on the maximum-entropy method.

$\langle P_4 \rangle = 0$. The increase in $\lambda_2$ and $\lambda_4$ implies a sharpening of the $f_{MEM}(\theta)$ function. For $\lambda_4 = 0$, the ODF in Eq. (18) is identical to the Bingham/Maier-Saupe distribution in Eq. (14), which was successfully applied in liquid crystals to describe the condensation of the nematic phase within a mean-field theory[42]. Herein, the comparison between the $f_{B/MS}(\theta)$ (or Eq. (18) with $\lambda_4 = 0$) and the experimental dataset of order parameters in Fig. 2 calls for a positive contribution of $\lambda_4$, especially in the $0.4 < \langle P_2 \rangle < 0.9$ region. In contrast, in the case of rod-like organic 4,4′-dimethoxyazoxybenzene compound, Humpries et al.[43] proposed an alternative model leading to an ODF similar to that in Eq. (18), with a negative contribution of $\lambda_4$. Moreover, these authors considered a fixed ratio between $\lambda_2$ and $\lambda_4$ in their model. For the clay minerals studied here, the correlation between $\lambda_2$ and $\lambda_4$ is achieved by first extracting the exact values of $(\lambda_2 - \lambda_4)$ pairs associated with each experimental $\langle P_2 \rangle - \langle P_4 \rangle$ pair and then by deriving the $\lambda_4 - \lambda_2$ relationship (see Supplementary Note 2, Supplementary Fig. 4). A good fit is obtained for $\lambda_4 = 0.005(\lambda_2)^5$, and the general ODF for clay platelets becomes:

$$f_{CM}(\theta) = k_{CM}(\lambda_2)\exp[\lambda_2 P_2(\cos\theta) + 0.005(\lambda_2)^5 P_4(\cos\theta)] \quad (21)$$

where $\lambda_2$ is a single parameter controlling the spreading parameter of the ODF and $k_{CM}(\lambda_2)$ is the normalization constant that satisfies Eq. (3). The index 'CM' refers to clay minerals, whereas $P_2(\cos\theta)$ and $P_4(\cos\theta)$ are the first two orders of the Legendre polynomial series reported in Eqs. (7) and (8), respectively. We show in Fig. 4 that the $f_{CM}(\theta)$ ODF based on the MEM well reproduces the correlation between the $\langle P_2 \rangle$ and $\langle P_4 \rangle$ order parameters for our experimental OD dataset. Such an excellent agreement does not guarantee, however, that the entire set of other $\langle P_l \rangle$ momenta are correctly accounted for by this new function. The adequacy of our model for the prediction of the $\langle P_6 \rangle$ order parameter (Fig. 4) provides, however, good indications that a cut-off of $n = 4$ in Eq. (17) is relevant and that $\lambda_i$ parameters for $i \geq 6$ are null. A crucial test for the validation of the $f_{CM}(\theta)$ function through its capacity to reproduce the full set of $\langle P_l \rangle$ order parameters is performed below on the basis of a direct comparison between the experimental and predicted ODF curves.

**Comparison between experimental and predicted orientation distribution functions**. The full validation of the general $f_{CM}(\theta)$ function for reproducing the entire signature of the experimental OD profiles is shown in Fig. 5 for representative engineered clay samples. To do so, the parameter $\lambda_2$ in the $f_{CM}(\theta)$ function is chosen so that the calculated $\langle P_2 \rangle$ value is equal to the experimental value. The experimental and calculated ODFs are then

compared without further adjustments (Fig. 5). The improvement provided by the $f_{CM}(\theta)$ function compared to standard models is also highlighted. This comparison is limited to the Fisher ($f_F(\theta)$), Bingham/Maier-Saupe ($f_{B/MS}(\theta)$), and mechanical compaction ($f_{MC}(\theta)$) distribution models, which provided the closest $\langle P_4 \rangle - \langle P_2 \rangle$ relations with respect to our experimental order parameters (Fig. 2). Here again, calculations are performed by adjusting the variable $m$ in Eqs. (12–14) to fit the experimental $\langle P_2 \rangle$ value. As expected from Fig. 2, all these standard models reproduce rather satisfactorily the experimental OD profile for $\langle P_2 \rangle = 0.2$ (Fig. 5a). For the Bingham/Maier-Saupe model at $\langle P_2 \rangle = 0.4$ (Fig. 5b), the calculated $\langle P_4 \rangle$ value is consistent with the measured value in Fig. 2. However, a systematic discrepancy is observed near $\theta = 0$ (or $\theta = \pi$) for higher degrees of anisotropy (Fig. 5c, d). Benefitting from the MEM theory, we attribute the excellent agreement between our experimental and predicted OD dataset to the consideration of the $\lambda_4$ contribution in the general $f_{CM}(\theta)$ ODF.

**Extension to natural clay-rich systems**. To go further, we investigate whether the proposed general ODF could be extended to the analysis of the anisotropy of clay platelet orientation in three complex natural systems, i.e., a shale, a soil crust, and a schist. These selected samples are representative of different clay-rich environments on Earth's surface or in the subsurface[1] where orientation of clay minerals is of particular interest. Preferred orientation of clay minerals in soils and shales plays indeed a key role on water and solutes migration[5–7,15,25] whereas anisotropy in clay particle orientation in schists is a marker of strains during metamorphism[44–46]. Compared to previous samples used to determine the shape of the general $f_{CM}(\theta)$ function, the organization of clay platelets in these three samples originates from extremely contrasted conditions in terms of depth, temperature, or pressure. Moreover, these samples display a complex assemblage of both clay and non-clay particles, as well as a wide range of particle sizes. Because of this complexity, the collected experimental XRS diagrams (Fig. 6) appear noisier than the monomineralic clay samples (Fig. 5). However, fitting the experimental ODFs (extracted from the *001* reflection of mica or illite clay minerals) with the general $f_{CM}(\theta)$ function leads to a very good agreement for the three natural samples investigated (Fig. 6). Such an agreement further confirms the ability of the new $f_{CM}(\theta)$ reference function to describe anisotropy in clay platelet orientation for complex natural media. The robustness of this function is evidenced for clay minerals formed under extremely different natural conditions and leading to a wide range of

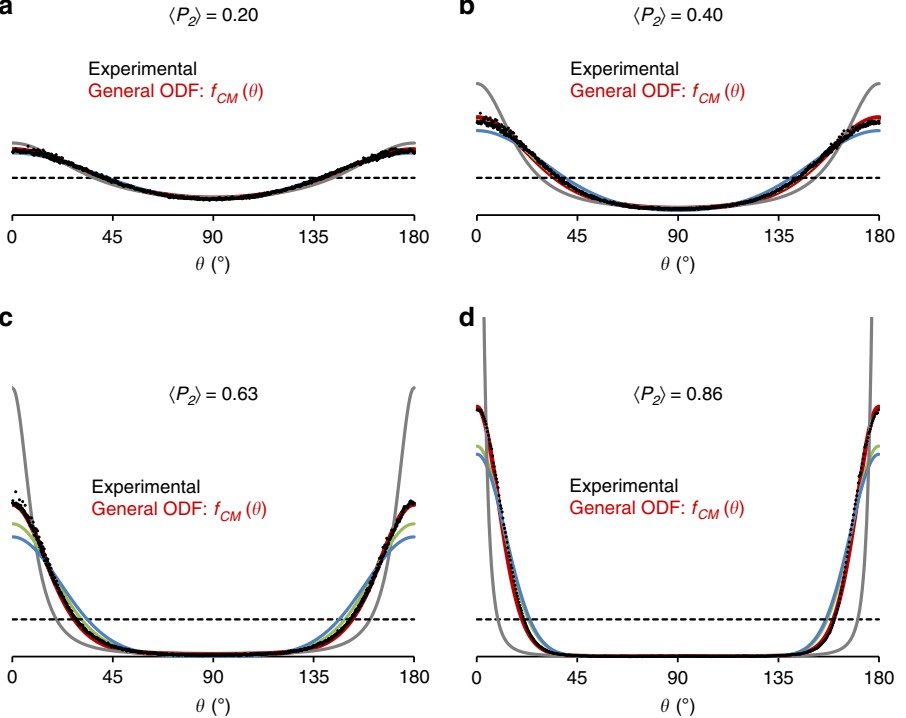

**Fig. 5** Validation of the general orientation distribution function for clay minerals. Comparison between experimental ODFs (black circles) and predicted ODFs (red lines) using the general orientation distribution function for clay minerals ($f_{CM}(\theta)$ using Eq. (21)). The selected samples display a gradual increase in anisotropy value with **a** compacted kaolinite with $\langle P_2 \rangle = 0.20$, **b** compacted kaolinite with $\langle P_2 \rangle = 0.40$, **c** centrifuged kaolinite with $\langle P_2 \rangle = 0.63$, and **d** sedimented smectite (fluorohectorite) with $\langle P_2 \rangle = 0.86$. Isotropic organization is shown as a horizontal dotted line for reference. Conventional distribution models are shown in blue, green, and gray for the Fisher ($f_F(\theta)$), Bingham/Maier-Saupe ($f_{B/MS}(\theta)$), and mechanical compaction ($f_{MC}(\theta)$) models, respectively.

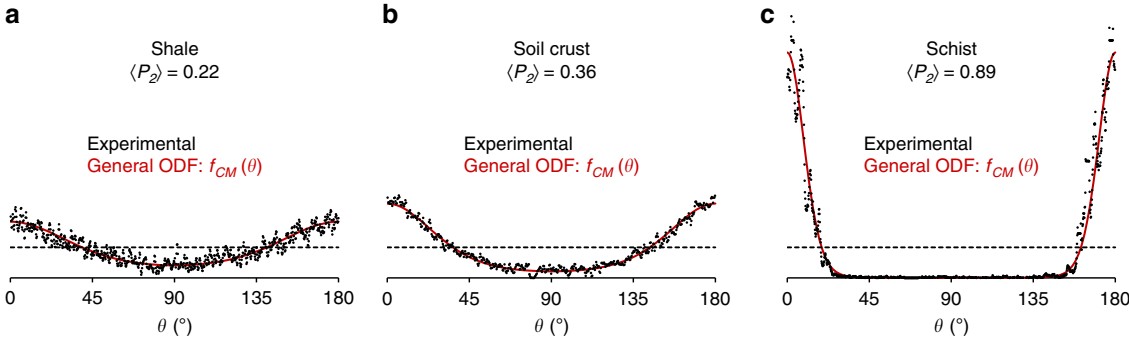

**Fig. 6** Application of the general orientation distribution function for natural clay-rich media. Experimental and best-fitted ODFs using the general $f_{CM}(\theta)$ ODF for clay minerals ($f_{CM}(\theta)$ using Eq. (21)) are shown as circles and red lines, respectively. The anisotropy in particle orientation described by the $\langle P_2 \rangle$ order parameter is extracted from the calculated $f_{CM}(\theta)$ functions. The samples are displayed with a gradual increase in anisotropy value with **a** a shale from Paris Basin with $\langle P_2 \rangle = 0.22$, **b** a soil crust from the Versailles experimental site with $\langle P_2 \rangle = 0.36$, and **c** a schist from the Armorican Hercynian belt with $\langle P_2 \rangle = 0.89$. Isotropic organization is shown as a horizontal dotted line for reference.

anisotropy degrees (i.e., for $\langle P_2 \rangle$ values ranging from 0.22 to 0.89; Fig. 6).

## Discussion
This work provides evidence for the existence of a reference function describing the ODF of clay platelets and is applicable for a large set of clay types, anisotropy ranges, and formation processes in both engineered and natural materials. Our methodology based on the MEM to derive this new numerical function contains, within experimental precision, all relevant information on the ODF (i.e., the full set of $\langle P_l \rangle$ order parameters) from the coupling between $\lambda_2$ and $\lambda_4$ parameters. By relating both entities, the unique general function $f_{CM}(\theta)$ in Eq. (21), with only one variable parameter, can be applied to describe the whole range of anisotropy degrees. Accordingly, this new function provides clear

improvements compared to conventional distribution models (Gaussian, Bingham/Maier-Saupe, etc.), which are limited to certain anisotropy domains. Furthermore, this general $f_{CM}(\theta)$ function paves the way for unifying the various global descriptors of the experimental OD profiles (e.g., maximum intensity or width of the distribution) used in the literature to interpret the preferred orientation of clay platelets.

One of the important aspects of this approach is that the $f_{CM}(\theta)$ function relies on the consideration of homogeneous samples. In the case of highly heterogeneous samples, the difference between experimental and predicted ODF should help in elucidating the coexistence of spatial regions (or particle populations) with contrasted anisotropic features[15,18]. Another important property of the $f_{CM}(\theta)$ function relies on its construction through the coupling between the $\lambda_2$ and $\lambda_4$ parameters. Here, a positive contribution of the $\lambda_4$ parameter is required to satisfy the $\langle P_4 \rangle - \langle P_2 \rangle$ relation for clay platelets, whereas a negative $\lambda_4$ contribution was considered in the case of rod-like organic compounds[43]. One may thus tentatively hypothesize that $(\lambda_2, \lambda_4)$ coupling may be somehow linked to the shape of the particle. Future work could thus aim at assessing the extent of the general function for other lamellar materials (e.g., graphene, MXenes, MoS$_2$, and layered double hydroxides)[47], as well as for other types of grain morphologies (tubes, rods, and wires) in an effort to generalize the methodology developed here to improve the design of engineered materials for the electronics, energy, or catalysis fields (e.g., ZnO[48], Co$_3$O$_4$[49], carbon nanotubes[50,51], halloysite[51], or imogolite[52,53]). Beyond this fundamental knowledge, a better description of the preferential orientation of clay platelets, as proposed here on the basis of the $f_{CM}(\theta)$ function, represents a key parameter for driving the design of new generations of innovative engineered materials in the field of packaging or high-temperature industrial applications with improved mechanical resistance and drastic fluid and gas permeability reduction[10–13].

We demonstrated here that the $f_{CM}(\theta)$ function is not only valid for pure clay-based samples but also mimics well the OD of clay platelets in three natural media (i.e., soil, shale, and schist). This result provides evidence for the applicability of this reference function to clay minerals in the case where these minerals are not the major mineralogical components. Accordingly, one may thus consider that this generic function represents a good proxy for the quantitative description of preferred orientation in the main different superficial environments of the Earth where clay minerals are found. In that regard, the $f_{CM}(\theta)$ function in Eq. (21) could be incorporated into current predictive models of caprock transport properties for oil, gas, or water resources. As far as clay barriers for waste repositories are concerned, the obtained diffusion coefficient for water can vary by a factor of 3–4 when considering either purely isotropic or perfectly anisotropic orientation of clay platelets[25]. Correctly accounting for the actual preferred orientation using the numerical function determined can be expected to considerably improve the predictive power of these transport models.

## Methods

**Samples**. Two types of sample series were considered in this study, i.e., monomineralic reference clay samples and clay-rich natural polymineralic samples. The selected reference clay minerals (i.e., kaolinite, mica, chlorite, vermiculite, and smectite) represent the most common different types of clay minerals in natural environments. Kaolinite KGa-2 with [(Al$_{7.60}$Ti$_{0.26}$Fe$^{3+}_{0.14}$)(Si$_{7.68}$Al$_{0.32}$)O$_{10}$(OH$_8$)] as the structural formula[54] was obtained from the Source Clays Repository of the Clay Minerals Society. Large monocrystals of mica clay minerals (i.e., muscovite) from the University of Poitiers collection were first sonicated to reduce particle size, and the <1 and 0.1–0.2 μm fractions were then collected by conventional centrifugation[55]. Chlorite sample from the T300 sample series of the Trimouns deposit (sample T340C, French Pyrenées) with (Al$_{2.27}$Mg$_{8.37}$Ca$_{0.20}$Na$_{0.14}$K$_{0.01}$Fe$^{2+}_{0.57}$Fe$^{3+}_{0.02}$)(Si$_{7.68}$Al$_{0.32}$)

O$_{20}$(OH$_{15.90}$ F$_{0.10}$)] as structural formula[56] were first micro-milled and the <2 μm clay fraction was then collected by conventional centrifugation[55]. Vermiculite samples with [(Mg$_{4.92}$Fe$^{2+}_{0.43}$Al$_{0.59}$Ti$_{0.04}$)(Si$_{5.66}$Al$_{2.34}$)O$_{20}$(OH$_4$)]Na$_{1.64}$ as the structural formula[57] and in the form of 0.1–0.2, 1–2, and 10–20 μm size fractions were obtained from a previous study[55]. Three types of smectite clay minerals considered here were obtained from previous studies. The first is a natural Wyoming MX80 montmorillonite, with the structural formula[58] [(Al$_{3.10}$Mg$_{0.56}$Fe$^{3+}_{0.18}$Fe$^{2+}_{0.16}$)(Si$_{7.96}$Al$_{0.04}$)O$_{20}$(OH$_4$)]Na$_{0.76}$. The two other smectites correspond to hydrothermally synthetized hydroxylated and fluorinated hectorite[59] endmembers with a generic structural formula [(Mg$_{5.2}$Li$_{0.8}$)(Si$_{8.0}$)O$_{20}$(OH,F)$_4$]Na$_{0.8}$. The three natural clay-rich polymineralogic samples selected (i.e., soil crust, shale and schist) correspond to environments with contrasted formation conditions where clay minerals are commonly found. The first sample is a centimeter-sized soil crust from the long-term experimental site in Versailles, France[60], with ~15% of <2 μm clay fraction. This crust is formed at the surface of the soil by soft sedimentation of clay particles (mainly mica-type and kaolinite) after rainwater. The second sample is a shale from the eastern Paris basin, France, extracted from the EST104 borehole at 453 m depth and belonging to the Callovo-Oxfordian formation[61]. This claystone contains approximately 25–50% clay minerals (mainly mica-type and smectite) resulting from an original sedimentation event followed by a compaction process under soft lithostatic temperature and pressure conditions. The third sample is a schist from Les Sables-d'Olonne, France, located in the south Armorican Hercynian belt[46]. This sample was formed under Barrowian metamorphic conditions corresponding to a folding process at a temperature up to 300 °C and leading to crystallization of mica-type clay minerals.

**Preparation of pure clay samples with different organizations**. Different preparation methods (compaction, centrifugation, and sedimentation) were considered to maximize the range of anisotropy degree in particle orientation for monomineralic engineered clay samples investigated here. All powders were first dried at 60 °C overnight and then sieved through <50 μm to limit the size of particle aggregates. The compaction method involved uniaxial compression of a given amount of powder in poly(tetrafluoroethylene) (PTFE) cylinders with a diameter of 0.64 cm. The centrifuged samples were obtained using the following procedure: dispersion of clay particles in water by sonication, successive centrifugation of the suspension directly in the PTFE cylinders under gravitational fields ranging from 4500 to 18,000 × g, and drying of the settled sample at 60 °C overnight. For the sedimentation procedure applied to the smectites that allowed thin, self-standing clay films to be obtained, the clay particles were first dispersed in water by sonication and then allowed to settle and dry at room temperature on a flat polymer substrate.

**Resin induration and sample slice cutting**. For resin impregnation, all self-standing samples were first dried overnight at 60 °C and placed in a cell under a primary vacuum to remove the water. Porosity was then filled with Methyl MethAcrylate (MMA) over a time span ranging from 3–14 days for monomineralic samples and up to 1 month for natural samples. The MMA resin was then polymerized in a 55 °C bath for at least 24 h using benzoyl peroxide (BPO) as a thermal initiator (BPO/MMA ratio of 0.5 wt.%[62]). The impregnated clay samples were then sawed perpendicularly to the main stratigraphic direction (i.e., longitudinally to the PTFE cylinders), and the thickness of the obtained lamellas was then reduced to ~500 μm by a successive polishing procedure before XRS analysis.

**X-ray scattering measurements**. The XRS experiments were carried out at the Laboratoire de Physique des Solides at Orsay, France. The X-ray beam is delivered by a copper rotating anode generator (RU H3R, Rigaku Corporation, Japan) and monochromatized ($\lambda_{CuK\alpha} = 1.5418$ Å) by a multilayer W/Si mirror (Osmic), providing a monochromatic beam of 600 × 600 μm² at the sample position. Two-dimensional XRS patterns were collected on a MAR345 detector (marXperts GmbH, Germany), with a 150 μm pixel size, placed at a sample-to-detector distance $D$ of 250 mm or 350 mm. The beam stop was placed 30 mm behind the sample. This setup allows us to reach a scattering vector modulus down to $Q_{min} = 0.2$ Å$^{-1}$ ($Q = 4\pi/\lambda \sin(\theta_B)$, where $\lambda$ is the incident wavelength and $2\theta_B$ is the scattering angle), i.e., $d$-spacing up to 31.5 Å ($d = 2\pi/Q$). The sample slices were mounted on a goniometer head so that the axis of the main stratigraphic direction was perpendicular to the incident X-ray beam. The assumption regarding the cutting of sample slices longitudinally to the main stratigraphic direction was verified on selected samples (see Supplementary Note 3, Supplementary Fig. 5). The ODFs were deduced from the angular modulation of intensity $I$ along the *001* diffraction ring (Eq. (4)) after background subtraction.

## Data availability

The data that support the findings of this study are available on request from the corresponding authors.

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

## Acknowledgements

The results presented are part of the Ph.D. thesis of T.D. granted by "Région Nouvelle-Aquitaine", University of Poitiers, France. Claude Veit (IC2MP, Poitiers, France) is thanked for the design and conception of the compaction cells for sample preparation. Dr. Folkert van Oort (ECOSYS, Versailles, France) and Prof. François Martin (GET, Toulouse, France) are thanked for providing the soil and chlorite samples, respectively. We acknowledge the CNRS interdisciplinary "défi Needs" through its "MiPor" program (Project TRANSREAC) and the European Union (ERDF) and "Région Nouvelle Aquitaine" for providing financial support for this study.

## Author contributions

F.H. and E.F. conceived the project. T.D., F.H., C.L., B.G., B.D. and E.T. designed and prepared the samples. T.D. and E.P. performed the XRS experiments. T.D., P.L., E.P. F.H., A.D. and E.F. analyzed the data and interpreted the results. All authors contributed to the discussion and to the writing or review of the paper.

## Competing interests

The authors declare no competing interests.
