## [Peer Review File · Nature Communications]

Reviewers' comments:

Reviewer #1 (Remarks to the Author):

This work deals with the orientation distribution (OD) function characterisation of clay minerals. The authors provide a general reference function of clay mineral alignment in a range of clay-rich engineered and natural materials. To my knowledge, this hasn't been done so far and the paper provides a detailed analysis of the different momenta of the investigated data and the authors managed to unravel the existence of a specific orientation signature independent of the type of clay platelet or preparation method chosen.

Preferred orientation is generally investigated via orientation distribution of crystallites by diffraction method and the anisotropy in clay platelet is usually illustrated by descriptors of the orientation distribution. The authors here opted for X-ray scattering (XRS) measurements and the results proved to be really solid and well presented.

The interpretation and conclusions are reasonable and are adequately supported and discussed, although there may be some inconsistencies (see specific comments). I personally would have chosen also Chlorite as a clay mineral given the ubiquitous nature of it. I warmly suggest to include also such a clay mineral in the set of investigated samples. Previous works have been done on the characterization of clay mineral assemblages for Paleoclimate and paleoenvironmental reconstructions and chlorite plays a very important role when considering the main four families of clay minerals (Smectite, Kaolinite, Illite and Chlorite)(Biscaye, 1965, Iacoviello et al. 2012, Franke and Ehrmann, 2011, Ehrmann et al, 2005, for example).

The reference orientation function has various potential applications whereas the preferred orientation consideration is needed. To my knowledge, to date, a reference function which describes the Orientation Distribution hasn't been shown. This paper helps in filling the gap resulting from the lack of reference function.

The English language is good for a non-native English reader, but the style may require some editing. The paper "fits" the Journal and is of sufficient weight to warrant publication. Therefore, I would recommend this paper for publication in Nature Communications, after the authors address specific comments and suggestions below:

ABSTRACT

Line 37: consider changing tense (identified and determined) in accordance with the previous tense.

Line 55: please clarify retention of what? (water?)

Line 59: remove or

Line 64: given their low cost

Line 65: and the availability

Line 66: remove the, orientation-control

Line 74: change maxima to maximum

Line 76: Bingham, please check throughout the manuscript

Line 88: remove a (Through "a" detailed)

Line 90: Change regardless to independent

Line 91: Change reproducing to "for the reproduction of"

RESULTS

Line 102: remove the (from "the" X-Ray...)

Line 106: change and verifies to , where:

Line 115: angle t should be italicised?

Line 141: write "order parameters"

Line 145: Hermans instead of Herman

Line 148: \cos must be italicised

Line 153-154: Please do provide reference. To my knowledge chlorite is also very common in both marine sediments and other environments (see Biscaye 1965).

Line 166: Change evidenced to evident

Line 191: change throughout to across

Line 194: change relations to relationships

Line 198-200: Colour choice is confusing to me. I suggest either changing model colour or adding a key also for these models so it will be really straightforward to link a dashed line to the relative model

Line 205: remove the (on "the" information)

Line 212: represents the Lagrange (instead of is)

Line 220: remove the (to "the" MEM)

Line 224: remove the ("the" maximum entropy theory)

Line 231: Fig. 2 calls instead of pleads

Line 233: et al. must be italicised

Line 254: Change relations to relationships

Line 279: Change Beneficiating to Benefitting

Line 284: natural systems, i.e. etc Please provide more info regarding the three systems. While I totally understand the first and second one, I am quite struggling to find a justification for the schist. Could you provide more background regarding this? I would suggest adding bentonite, since it could prove to be an optimal system for various applications, from medical to pottery, to purification and groundwater barrier for radioactive waste management

Line 291: 001 must be italicised

Line 328: other - other? other? To my knowledge, they are not technically single-layered materials like graphene.

Please clarify.

Line 338: mimics well

line 358: is that Fe 2+ or 3+? please specify

Line 375: leave a space between 300 and C

Line 384: 4500 and 18000 are RPM or G? please confirm

REFERENCE

Line 411: It looks like 2 different approaches have been used: 1 -all authors were reported; 2- et al. was written instead of reporting all the authors' list. Keep it consistent

Line 421: Check CO₂ and H₂

Line 443: Geophysics shouldn't be upper case

Nothing to add regarding the Supplementary Information.

Best Regards,

Francesco Iacoviello

Reviewer #2 (Remarks to the Author):

Dear Nature Commun. editor,

I would preface my review by noting that my expertise is primarily in clay science and nanogeochemistry and that I have not delved deeply into the orientational distribution functions discussed in the manuscript. Within my area of expertise, the work presented by Dabat and coworkers is scientifically sound and novel and the quality of the research is impressive. The overall design of the project is logical and draws on an appropriate combination of wet-chemical and novel X-ray scattering experiments carried out on both pure and environmental samples, combined with theory and a review of existing approaches. The text is well written overall but would benefit from mild editing for style. My only other (mild) comment is that the sentence on lines 71-72, which discusses how preferential orientation is 'generally studied', should probably cite references from more than one research group. Overall, I would recommend that the manuscript be accepted for publication.

Best wishes,

Ian C. Bourg

Princeton University

Reviewer #4 (Remarks to the Author):

This manuscript presents original measurements obtained by X-ray diffraction to estimate the orientational distribution function of clay-rich media. The paper is easy to follow and sounds solid scientifically. Knowing the orientational distribution function is important for many applications (polymers, liquid crystals as well) since the macroscopic properties depend often on the structural anisotropy of materials. I think the science is deep and would probably be appealing for the audience of nature communication.

However the manuscript suffers from few weaknesses that the authors should correct:

-the sentence "Because the Legendre polynomials are odd functions for odd l values, the sum can be restricted to even values of l ." does not explain why the odd terms are null. The odd terms are null because there is no polar ordering of the platelets. A platelet has the same chance to be oriented by $+\theta$ or $-\theta$.

-I think that the authors could be more precise about why X-ray gives the full distribution function $f(\theta) = \sum P_l(\theta)$ with l even. They could mention also why do they stop at P_6 ? I had the impression that X-ray gives the full function. If terms above P_4 are negligible this should be stated but figure 2 seems to show that P_6 is not negligible.

I had hopes once I start reading the manuscript that the authors would explain why X-ray is much better than Raman or IR (two techniques for measuring p_2 and p_4 for example) but this has not been the case. X-ray, to the best of my knowledge, gives directly $f(\theta)$ that can then be fitted with a

function that includes $P(\theta)$ and the goal is to find the P which is not that easy because several sets of parameters can be solutions.

Furthermore I don't understand why the authors dissociates P_4-P_2 and P_6-P_2 . I would have hope to see more distribution functions as shown in Fig1C but was disappointed to see only binary diagrams (p_2-p_4 , p_6-p_2) or simply orientational functions with varying P_2 like if P_4 and P_6 were not of interest anymore.

Because of this aspect, my interest faded away. I would have like to see 3D diagram $f(\theta)=f(P_2,P_4,P_6)$ to evaluate the coexistence diagrams of these 3 parameters, for example. The entropy model could be also generalized with 3 parameters : λ_2 , λ_4 and λ_6 .

Here again the authors stopped moving their approach deeper.

For this main reason, I think that additions are necessary to provide clear cut of the value of the present work in comparison with what has been done already in the field of orientation function distribution.

This is yet a very nice work with a lots of promises and the elements above aim to improve the quality of this manuscript.

ANSWERS TO REVIEWERS

REVIEWER #1

General comments

This work deals with the orientation distribution (OD) function characterisation of clay minerals. The authors provide a general reference function of clay mineral alignment in a range of clay-rich engineered and natural materials. To my knowledge, this hasn't been done so far and the paper provides a detailed analysis of the different momenta of the investigated data and the authors managed to unravel the existence of a specific orientation signature independent of the type of clay platelet or preparation method chosen.

Preferred orientation is generally investigated via orientation distribution of crystallites by diffraction method and the anisotropy in clay platelet is usually illustrated by descriptors of the orientation distribution. The authors here opted for X-ray scattering (XRS) measurements and the results proved to be really solid and well presented.

The interpretation and conclusions are reasonable and are adequately supported and discussed, although there may be some inconsistencies (see specific comments). I personally would have chosen also Chlorite as a clay mineral given the ubiquitous nature of it. I warmly suggest to include also such a clay mineral in the set of investigated samples. Previous works have been done on the characterization of clay mineral assemblages for Paleoclimate and paleoenvironmental reconstructions and chlorite plays a very important role when considering the main four families of clay minerals (Smectite, Kaolinite, Illite and Chlorite)(Biscaye, 1965, Iacoviello et al. 2012, Franke and Ehrmann, 2011, Ehrmann et al, 2005, for example).

The reference orientation function has various potential applications whereas the preferred orientation consideration is needed. To my knowledge, to date, a reference function which describes the Orientation Distribution hasn't been shown. This paper helps in filling the gap resulting from the lack of reference function.

The English language is good for a non-native English reader, but the style may require some editing. The paper "fits" the Journal and is of sufficient weight to warrant publication. Therefore, I would recommend this paper for publication in Nature Communications, after the authors address specific comments and suggestions below:

According to the comment raised by the referee regarding adding chlorite to the study we decided to find this new sample and perform the analyses. The results of the 14 additional experiments show that the reference orientation function also works very well for chlorite. We would like to thank the reviewer for his suggestion as we are all convinced that it will help to broaden the readership of the paper.

The analyses on chlorite were thus added to the dataset and the Figs. 2 and 4, as well as Fig. S4 were changed accordingly. Finally, we added a sentence in the introduction regarding interest of clay mineral analysis for paleoenvironmental reconstruction together with 2 new references. The sentence added is (L50-51): *"In sedimentary rocks such as mudrocks or shales, clay minerals are common markers for paleoenvironmental reconstructions^{3,4} ..."*.

ABSTRACT

Line 37: consider changing tense (identified and determined) in accordance with the previous tense.

Corrected in the revised version of the manuscript.

Line 55: please clarify retention of what? (water?)

We modified the sentence into “on retention and anisotropic transfer of water and solutes” to make it clear.

Line 59: remove or

Corrected in the revised version of the manuscript.

Line 64: given their low cost

Corrected in the revised version of the manuscript.

Line 65: and the availability

Corrected in the revised version of the manuscript.

Line 66: remove the, orientation-control

Corrected in the revised version of the manuscript.

Line 74: change maxima to maximum

Corrected in the revised version of the manuscript.

Line 76: Bingham, please check throughout the manuscript

Corrected in the revised version of the manuscript. No other occurrence of this mistake was detected in the text.

Line 88: remove a (Through "a" detailed)

Corrected in the revised version of the manuscript.

Line 90: Change regardless to independent

Corrected in the revised version of the manuscript.

Line 91: Change reproducing to "for the reproduction of"

Corrected in the revised version of the manuscript.

RESULTS

Line 102: remove the (from "the" X-Ray...)

Corrected in the revised version of the manuscript.

Line 106: change and verifies to , where:

Corrected in the revised version of the manuscript.

Line 115: angle t should be italicised?

Corrected in the revised version of the manuscript.

Line 141: write "order parameters"

We modified the sentence into “*The different momenta of the distribution function, also defined as order parameters, are given by:*” to make it clearer.

Line 145: Hermans instead of Herman

Corrected in the revised version of the manuscript.

Line 148: \cos must be italicised

Corrected in the revised version of the manuscript.

Line 153-154: Please do provide reference. To my knowledge chlorite is also very common in both marine sediments and other environments (see Biscaye 1965).

In the revised version of the manuscript the reference to Ito and Wagai (2017) and Biscaye (1965), already cited in the introduction, were added here.

Line 166: Change evidenced to evident

Corrected in the revised version of the manuscript.

Line 191: change throughout to across

Corrected in the revised version of the manuscript.

Line 194: change relations to relationships

Corrected in the revised version of the manuscript.

Line 198-200: Colour choice is confusing to me. I suggest either changing model colour or adding a key also for these models so it will be really straightforward to link a dashed line to the relative model

As suggested by the reviewer we added a label in the revised version of Fig. 2 to highlight the different models. The caption of Fig. 2 was modified accordingly.

Line 205: remove the (on "the" information)

Corrected in the revised version of the manuscript.

Line 212: represents the Lagrange (instead of is)

Corrected in the revised version of the manuscript.

Line 220: remove the (to "the" MEM)

Corrected in the revised version of the manuscript.

Line 224: remove the ("the" maximum entropy theory)

Corrected in the revised version of the manuscript.

Line 231: Fig. 2 calls instead of pleads

Corrected in the revised version of the manuscript.

Line 233: et al. must be italicised

Corrected in the revised version of the manuscript.

Line 254: Change relations to relationships

Corrected in the revised version of the manuscript.

Line 279: Change Beneficiating to Benefitting

Corrected in the revised version of the manuscript.

Line 284: natural systems, i.e. etc Please provide more info regarding the three systems. While I totally understand the first and second one, I am quite struggling to find a justification for the schist. Could you provide more background regarding this? I would suggest adding bentonite, since it could prove to be an optimal system for various applications, from medical to pottery, to purification and groundwater barrier for radioactive waste management

We understand that the original version was slightly confusing for the reader regarding the choice of the 3 natural samples. We modified the text in order not to mean that the 3 samples are the most representative natural clay-rich environments but rather to indicate that these samples are typical environments where analysis of preferred orientation of clay minerals is of interest for several aspects. Accordingly, the use of the schist sample is justified by the fact that the analysis of preferred orientation in such type of rock is commonly used as a marker of strains that occurred during the deformation history of the clay material. In the revised version we modified the paragraph and added two references concerning the schist. The new version is now (L294-298):
“These selected samples are representative of different clay-rich environments on Earth’s surface or in the subsurface¹ where orientation of clay minerals is of particular interest. Preferred orientation of clay minerals in soils and shales plays a key role on water and solutes migration^{3-5,20} whereas anisotropy in clay particle orientation in schists is a marker of strains during metamorphism³⁸⁻⁴⁰.”

Concerning referee’s remark about bentonite, we agree that the analysis of preferred orientation in designed bentonite-based materials is of interest for various applications. However these materials

are prepared after milling and re-processing of the original bentonite and the final preferred orientation of clay minerals in these artificial systems is no more connected to the original natural bentonite bed. For this reason we decided not to include a rock extracted from a natural bentonite bed in the dataset. Note however that the MX-80 montmorillonite used in this study was extracted from a natural bentonite bed and can be considered as a reference in the context of artificial bentonite-based materials evoked by the reviewer.

Line 291: 001 must be italicised

Corrected in the revised version of the manuscript.

Line 328: other - other? other? To my knowledge, they are not technically single-layered materials like graphene. Please clarify.

To avoid confusion between single or multiple-layered materials we changed the sentence into “for other lamellar materials”

Line 338: mimics well

Corrected in the revised version of the manuscript.

Line 358: is that Fe 2+ or 3+? please specify

Iron is considered as Fe²⁺. This is now corrected in the revised version of the manuscript. We added the reference to the work of Marcos et al. (2003) containing the original chemical analysis of the sample and on which the study of Reinholdt et al. (2013) was based.

Line 375: leave a space between 300 and C

Corrected in the revised version of the manuscript.

Line 384: 4500 and 18000 are RPM or G? please confirm

The standard unity is a multiple of gravitational field (g and not G as in the original version). This is now better defined in the revised version.

REFERENCE

Line 411: It looks like 2 different approaches have been used: 1 -all authors were reported; 2- et al. was written instead of reporting all the authors' list. Keep it consistent

We followed the Nat. Com. guidelines regarding reference list of authors which indicate that “*All authors should be included in reference lists unless there are six or more, in which case only the first author should be given, followed by 'et al.'*”. No change was made in the revised version.

Line 421: Check CO₂ and H₂

Corrected in the revised version of the manuscript.

Line 443: Geophysics shouldn't be upper case

Corrected in the revised version of the manuscript.

Nothing to add regarding the Supplementary Information.

We are very grateful again for the time given to provide detailed and constructive comments on this manuscript.

Best Regards,

Francesco Iacoviello

REVIEWER #2

General comment

I would preface my review by noting that my expertise is primarily in clay science and nanogeochemistry and that I have not delved deeply into the orientational distribution functions discussed in the manuscript. Within my area of expertise, the work presented by Dabat and coworkers is scientifically sound and novel and the quality of the research is impressive. The overall design of the project is logical and draws on an appropriate combination of wet-chemical and novel X-ray scattering experiments carried out on both pure and environmental samples, combined with theory and a review of existing approaches. The text is well written overall but would benefit from mild editing for style. My only other (mild) comment is that the sentence on lines 71-72, which discusses how preferential orientation is 'generally studied', should probably cite references from more than one research group. Overall, I would recommend that the manuscript be accepted for publication.

Best wishes,

Ian C. Bourg

Princeton University.

We thank the referee for the positive comments about this work. The referee is right that the sentence about how ODF are generally studied was restricted to one research group. In the revised version we added a reference to another group in Germany (Dohrmann et al. 2009) as well as to one of our study (cited elsewhere in the text; Hubert et al., 2013). We also added references to the works of Cebula et al. (1979) and Hall et al. (1983) in order to present earlier works about ODF measurements on clay minerals by diffraction techniques.

Concerning possible mild English improvement, we are as non-English native fully aware about our limitations on this point. That is the reason why the manuscript was sent to "AJE: American Journal Experts", a freelance editor, prior its original submission, for improvement of English grammar, clarity, and consistency.

REVIEWER #4

General comment

This manuscript present original measurements obtained by Xray diffraction to estimate the orientational distribution function of clay-rich media. The paper is easy to follow and sounds solid scientifically. Knowing the orientational distribution function is important for many applications (polymers, liquid crystals as well) since the macroscopic properties depends often on the structural anisotropy of materials. I think the science is deep and would probably be appealing for the audience of nature communication. However the manuscript suffers from few weaknesses that the authors should correct:

1- the sentence “Because the Legendre polynomials are odd functions for odd l values, the sum can be restricted to even values of l .” does not explain why the odd terms are null. The odd terms are null because there is no polar ordering of the platelets. A platelet has the same chance to be oriented by $+\theta$ or $-\theta$.

We agree with the reviewer on the fact that the null values for odd terms of the Legendre polynomials originate from the symmetry of the $f(\theta)$ function given by Eq. (2): $f(\theta) = f(\pi - \theta)$. To avoid confusion we made it more explicit and the sentence was modified in the revised version into (L127-128): “*The sum can be restricted to even values of l as Eq. (2) leads to null values of odd terms of the series.*”.

2- I think that the authors could be more precise about why X rays brings the full distribution function $f(\theta) = \sum P_l(\theta)$ with l even. They could mention also why do they stop at P_6 ? I had the impression that $f(\theta)$ Xray gives the full function. If terms above P_4 are negligible this should be stated but figure 2 seems to show that p_6 is not negligible.

It can be seen from Eq. (4) that XRS analysis provides the full distribution function $f(\theta)$, with no assumption on its development in terms of the Legendre polynomials and Eq. (10) shows that the $\langle P_l \rangle$ values can be calculated directly from the experimental $f(\theta)$ distribution function, for all values of l . In the revised version of the manuscript this is now more explicit with the addition of the sentence (L141-142): “*The actual “fingerprint” of the shape of the ODF and the whole set of momenta $\langle P_l \rangle$ are accessible from XRS analysis according to Eqs. (4) and (10).* This sentence is followed by a sentence describing the advantage of XRS compared to Raman or IR techniques (see our answer to comment #3 from the referee).

Fig. 2 should not suggest that $\langle P_l \rangle$ values $>P_6$ are null. The question at this point of the manuscript is to focus on the 3 first momenta just to (i) highlight the presence of common signature for different clay minerals and (ii) show the limit of conventional functions. We changed the sentence (L157-159) to make it clear that our data analysis is illustrated only on the 3 first momenta of the full $\langle P_l \rangle$ series: “*Among the whole set of $\langle P_l \rangle$ order parameters accessible by XRS analysis, Fig. 2 shows the first momenta $\langle P_4 \rangle$ and $\langle P_6 \rangle$ as a function of $\langle P_2 \rangle$ for all samples investigated (with ODF determined from Eq. (4) and order parameters from Eq. (10)) to assess a potential common signature of ODFs.*”.

3- I had hopes once I start reading the manuscript that the authors would explain why Xray is much better than Raman or IR (two techniques for measuring p_2 and p_4 for example) but this has not been the case. Xray, to the best of my knowledge, gives directly $f(\theta)$ that can then be fitted with a function that includes $P_l(\theta)$ and the goal is the find the P_l which is not that easy because several sets of parameters can be solutions.

In L141-145 of the revised version, the advantage of XRS compared to IR or Raman is now detailed just after mentioning that XRS provide the full shape of the ODF: “*The actual “fingerprint” of the shape of the ODF and the whole set of momenta $\langle P_l \rangle$ are accessible from XRS analysis according to Eqs. (4) and (10). This contrasts with other vibrational techniques such as infrared or Raman spectroscopies, which provide only $\langle P_2 \rangle$ or $\langle P_2 \rangle$ and $\langle P_4 \rangle$ momenta^{32,33}, respectively, thus limiting the number of Legendre polynomials to the fourth rank order in Eq. (5).*”

Concerning the second sentence of referee’s comment (which seems also to concern the other following comments) one should discriminate the description of the $f(\theta)$ (i) as a sum of Legendre polynomials given by Eq. (5) from (ii) the exponential function given by the MEM theory as Eq. (17):

-As a Legendre series, the ODF (or any type of function) can be described at the condition that all $\langle P_l \rangle$ values are considered. This is not a real “fit” but the solution that satisfies Eq. (5) just requires the entire set of P_l values. Without considering all P_l values, Eq. (5) is, by definition, inoperable to reconstruct all types of ODF especially at high degree of anisotropy.

-In the case of the MEM theory, the exponential expression of the $f_{MEM}(\theta)$ account for all P_l , irrespective of the chosen number of λ parameters. Compared to Eq. (5) the benefit of the MEM approach is that there is no truncation in the description of the different momenta of the distribution.

Because the manipulation of the set of P_l in Eq. (5) is not an easy task (as also indicated by the reviewer) the literature is full of consideration of simple reference functions (Gaussian, Lorentzian, etc...). These functions “correlate” the different P_l of the distribution. In our study, the MEM allows defining a new reference function, that also generates the full set of P_l , based on a minimum number of parameters λ_i that are constrained by the first P_l of the distribution. We added a sentence (L217-223) in the revised version in the section devoted to the MEM description in order to make it clear for the reader: “*The MEM method has been proven to be very efficient for reconstructing the full orientation function when only a few order parameters are known^{38,39}. Indeed such a reduced knowledge yields Eq. (5) inoperable for high degree of anisotropy. The situation differs here as XRS analysis provides the full $f(\theta)$ ODF and thus the entire set of order parameters. The objective in the present work is to benefit from MEM theory to derive a reference distribution function based on a minimum number of parameters (i.e., λ_i coefficients) so that the whole set of $\langle P_l \rangle$ momenta would be automatically accounted for.*”

We hope that this addition will avoid confusion regarding the need or not of different P_l values to define a distribution function.

4- Furthermore I don’t understand why the authors dissociates P_4 - P_2 and P_6 - P_2 . I would have hope to see more distribution functions as shown in Fig1C but was disappointed to see only binary diagrams (p_2 - p_4 , p_6 - p_2) or simply orientational functions with varying P_2 like if P_4 and P_6 were not of interest anymore. Because of this aspect, my interest faded away. I would have like to see 3D diagram $f(\theta)=f(P_2,P_4,P_6)$ to evaluate the coexistence diagrams of these 3 parameters, for example. The entropy model could be also generalized with 3 parameters: λ_2 , λ_4 and λ_6 . Here again the authors stopped moving their approach deeper.

Again, the choice of selecting only the first momenta of the distribution (P_4 and P_2 in Fig. 3 and Fig. 4) when dealing with MEM theory is related to the fact that, as mentioned above, the exponential form of the $f_{MEM}(\theta)$ function makes the whole set of P_l values taken into account. There is no truncation of the Legendre series such as indicated in Eq. 5. We hope that the above-

mentioned addition (see answer to the previous comment) in the revised version will help the reader on these theoretical aspects.

From a practical viewpoint, the fMEM(theta) function can be based on the consideration of only lambda_2, or of lambda_2 + lambda_4, or of lambda_2 + lambda_4 + lambda_6, or of lambda_2 + ... + lambda_n. When only considering lambda_2 it is rather straightforward to show that Eq. 17 degenerates into Eq. 14 corresponding to the Birgham/Maier Saupe function. This is indicated L239 of the revised version “*For $\lambda_4=0$, the ODF in Eq. (18) is identical to the Bingham/Maier-Saupe distribution in Eq. (14)...*”. Again this function contains all P1 momenta of the distribution and the MEM theory implies that higher P1 parameters are correlated (roughly said as in the most “entropic” configuration) to the P1-2 parameters. Because Fig. 2 shows that for a given P2 value this function does not allow reproducing the P4/P2 relation, there is not real need to go further for higher order parameters. The situation differs when considering lambda_2 + lambda_4 as in Eq. 18. As shown here by adjusting the relation in the couple (lambda_2, lambda_4) it is possible to fit the P4/P2 relation. An important point is that, because it is possible to reproduce the P6/P2 relation only by considering lambda_2 and lambda_4, the lambda_6 parameter can be taken equal to zero. In other word this would end up to an Eq. 17 with all lambda_n for n>4 equal to zero. Not going on the consideration of lambda_6 is not really a question of moving or not the MEM approach deeper but is rather based on the fact that the goal of the MEM theory is, by definition and as written at L206 to find the “*most probable ODF that is consistent with a set of few known $\langle P_i \rangle$ values*”. Accordingly there is no need to over generalise a MEM model in Eq. 17 to end up with null values for other lambda_n parameters. We modified a sentence L259-260 in the revised version to emphasize that lambda_6 is not necessary: “*The adequacy of our model for the prediction of the $\langle P_6 \rangle$ order parameter (Fig. 4) provides, however, good indications that a cut-off of n=4 in Eq. (17) is relevant and that lambda_i parameters for i>=6 are null.*”. We hope that it will help the reader and the referee about why there is no need to go further than P4 or P6 value when building the fMEM(theta) function. Moreover all attempts of plotting 3D diagrams f(theta)=f(P2,P4,P6) as suggested by the reviewer showed that these latter are less clear to assess the validity of the fMEM(theta) function compared to the presentation of very basic but efficient P4/P2 and P6/P2 relations. Finally, it is important to point out that, once the fMEM(theta) is defined based on the minimum number of information (i.e., through the P4/P2 relation), the full validation of the obtained function is obtained through the comparison between experimental and predicted ODFs. This is indicated L260-262 as: “*A crucial test for the validation of the $f_{CM}(\theta)$ function through its capacity to reproduce the full set of $\langle P_i \rangle$ order parameters is performed below on the basis of a direct comparison between the experimental and predicted ODF curves.*”.

For this main reason, I think that additions are necessary to provide clear cut of the value of the present work in comparison with what has been done already in the field of orientation function distribution. This is yet a very nice work with a lots of promises and the elements above aim to improve the quality of this manuscript.

We thank the referee for the positive comments about this work. We are convinced that the additions/modifications/clarifications brought to the revised version of the text, in light with the constructive comments raised by the referee, will improve the quality of the manuscript.

REVIEWERS' COMMENTS:

Reviewer #1 (Remarks to the Author):

Dear Authors,

I have found the paper much improved since the last time I reviewed it. I am fully satisfied with the work carried on by the Authors. All my concerns were fully addressed and I am now happy to recommend it for publication without any further amendments.

I do believe that the changes could enhance the readership, broadening the audience of the work. In particular, I think that the addition of chlorite could also make the paper more appealing to the paleoclimate clay mineral community. I am really happy to see that the Authors took this suggestion into consideration and glad at the same time that the 14 additional experiments show that the reference orientation function also works very well for chlorite.

Best Regards,

Dr Francesco Iacoviello

Reviewer #4 (Remarks to the Author):

I am satisfied by the responses and the associated format and scientific corrections that have been done in this revised version.

I believe this is a nice and complete work that deserves publication in Nature Communications. This work meet all criteria for publication in a high impact journal.